# l-Theanine Protects Bladder Function by Suppressing Chronic Sympathetic Hyperactivity in Spontaneously Hypertensive Rat

**DOI:** 10.3390/metabo11110778

**Published:** 2021-11-14

**Authors:** Kanako Matsuoka, Hidenori Akaihata, Junya Hata, Ryo Tanji, Ruriko Honda-Takinami, Akifumi Onagi, Seiji Hoshi, Tomoyuki Koguchi, Yuichi Sato, Masao Kataoka, Soichiro Ogawa, Yoshiyuki Kojima

**Affiliations:** Department of Urology, Fukushima Medical University School of Medicine, 1 Hikarigaoka, Fukushima 960-1295, Japan; kanaco@fmu.ac.jp (K.M.); hakai@fmu.ac.jp (H.A.); akju826@fmu.ac.jp (J.H.); tanji422@fmu.ac.jp (R.T.); ruriko-t@fmu.ac.jp (R.H.-T.); onaji@fmu.ac.jp (A.O.); uro-hosi@fmu.ac.jp (S.H.); gucchii@fmu.ac.jp (T.K.); ysato@fmu.ac.jp (Y.S.); masaoka@fmu.ac.jp (M.K.); soh@fmu.ac.jp (S.O.)

**Keywords:** l-theanine, chronic sympathetic hyperactivity, bladder dysfunction, prevention, oral administration

## Abstract

Chronic sympathetic hyperactivity is known to affect metabolism and cause various organ damage including bladder dysfunction. In this study, we evaluated whether l-theanine, a major amino acid found in green tea, ameliorates bladder dysfunction induced by chronic sympathetic hyperactivity as a dietary component for daily consumption. Spontaneously hypertensive rats (SHRs), as an animal model of bladder dysfunction, were divided into SHR–water and SHR–theanine groups. After 6 weeks of oral administration, the sympathetic nervous system, bladder function, and oxidative stress of bladder tissue were evaluated. The mean blood pressure, serum noradrenaline level, and media-to-lumen ratio of small arteries in the suburothelium were significantly lower in the SHR–theanine than in the SHR–water group. Micturition interval was significantly longer, and bladder capacity was significantly higher in the SHR–theanine than in the SHR–water group. Bladder strip contractility was also higher in the SHR–theanine than in the SHR–water group. Western blotting of bladder showed that expression of malondialdehyde was significantly lower in the SHR–theanine than in the SHR–water group. These results suggested that orally administered l-theanine may contribute at least partly to the prevention of bladder dysfunctions by inhibiting chronic sympathetic hyperactivity and protecting bladder contractility.

## 1. Introduction

Green tea is a popular beverage in Japan. Consumption of green tea extract has also been reported to provide various benefits related to health, including its benefits for weight reduction [1], antiviral therapy [2], and even cancer treatment [3,4]. Most such research has referred to the benefits of polyphenol-enriched tea extracts such as catechins. However, increasing attention in recent years has focused on l-theanine, a unique amino acid contained in green tea. Since l-theanine crosses the blood–brain barrier [5], its psychological effects have been studied [6,7]. For instance, l-theanine increases alpha-wave activity, indicating effects in increasing relaxing and reducing anxiety [8]. l-theanine is expected to exert these psychological effects [9], but also physical effects such as reduction of sympathetic overactivity [10] and oxidative stress [11].

Elderly individuals are more likely to show lower urinary tract symptoms (LUTS) such as frequent urination and urinary incontinence. LUTS can be divided into bladder storage symptoms and voiding symptoms, and patients presenting with both types of symptoms simultaneously are not uncommon. For instance, bladder dysfunction associated with chronic bladder ischemia is assumed to cause bladder storage symptoms via the sensitization of afferent pathways in mild cases, and voiding symptoms due to detrusor underactivity of the bladder in severe cases [12]. The pathophysiology of bladder dysfunction is multifactorial, and it includes chronic bladder ischemia and sympathetic nervous system overactivity, which are reduced by l-theanine. We thus considered that l-theanine may effectively prevent bladder dysfunction, because sympathetic nerve hyperactivity and oxidative stress have been considered as risk factors for bladder dysfunction, and are suppressed by l-theanine [13,14,15].

The spontaneously hypertensive rat (SHR) was developed as an animal model of genetic hypertension [16] and is also used as a model of frequent urination [17,18]. Increased urinary frequency and bladder dysfunction in SHR are assumed to be related to the prolonged sympathetic nerve hyperactivity and chronic bladder ischemia caused by hypertension. As a result, the SHR appears suitable as an animal model to investigate any preventive effects of l-theanine on bladder dysfunction caused by chronic hypertension.

In this study, to reveal the utility of l-theanine as a preventive food against bladder dysfunction, we examined whether l-theanine protects bladder function in SHR.

## 2. Results

The general features of SHRs after 6 weeks of treatment in both groups are shown in Table 1. Daily fluid intake, body weight, and bladder weight did not differ significantly between groups. HR did not differ significantly between groups. SBP and DBP tended to be lower in the SHR–theanine group than in the SHR–water group, but the differences were not significant. On the other hand, MBP after 6 weeks of treatment was significantly lower in the SHR–theanine group than in the SHR–water group (*p* = 0.046). Among serum catecholamine concentrations, serum dopamine concentration did not differ significantly between groups (*p* = 0.937; Figure 1A), serum noradrenaline concentration was significantly decreased in the SHR–theanine group compared with the SHR–water group (*p* = 0.041; Figure 1B), and serum adrenaline concentration tended to be lower in the SHR–theanine group than in the SHR–water group (*p* = 0.065; Figure 1C).

### 2.1. Voiding Behaviors in SHRs after 6 Weeks

Figure 2 shows the results of metabolic cage experiments. The 24 h urine volume and daytime frequency were almost equal between groups (Table 2). Both 24 h micturition frequency (*p* = 0.004) and nighttime frequency (*p* = 0.007) were significantly decreased in the SHR–theanine group compared with the SHR–water group. Mean voided volume was significantly greater in the SHR–theanine group than in the SHR–water group (*p* = 0.043). These results indicate that l-theanine attenuated decreases in single voided volume in SHRs and prevented frequent urination, particularly at night.

### 2.2. Cystometric Parameters in Conscious Rats

The results of cystometric analyses are shown in Table 3. Representative cystometrograms from both groups are shown in Figure 3. Intercontractile interval was significantly longer in the SHR–theanine group than in the SHR–water group (*p* = 0.002). Voided volume (*p* = 0.029) was significantly greater in the SHR–theanine group than in the SHR–water group. No significant differences in baseline pressure, residual urine volume, maximum pressure, or bladder compliance were evident between groups. These results indicate that l-theanine may prevent lower urinary tract dysfunction in SHR.

### 2.3. Assessment of Bladder Contractile Strength

Bladder strip contractility in response to various stimuli was compared between groups. Mean contractile response to 80 mM KCl was significantly higher in the SHR–theanine group than in the SHR–water group (*p* = 0.010; Figure 4A). The SHR–theanine group showed a significantly higher mean contractile response induced by EFS from 1 Hz to 32 Hz than the SHR–water group (1 Hz, *p* = 0.001; 2 Hz, *p* = 0.004; 4 Hz, *p* = 0.010; 8 Hz, *p* = 0.010; 16 Hz, *p* = 0.008; 32 Hz, *p* = 0.016) (Figure 4B). Mean contractile responses to 1 mM ATP (*p* = 0.006) and carbachol at concentrations from 100 nM to 1 mM (1 nM, *p* = 0.512; 10 nM, *p* = 0.173; 100 nM, *p* = 0.043; 1 µM, *p* = 0.043; 10 µM, *p* = 0.002; 100 µM, *p* = 0.004; 1 mM, *p* = 0.005) were significantly higher in the SHR–theanine group than in the SHR–water group (Figure 4C,D). These results indicate that l-theanine increased bladder contractility in SHR of the same weeks old.

### 2.4. Expression of Oxidative Stress Markers in the Bladder 

Protein expression of MDA, a marker of oxidative stress, was analyzed by Western blotting and immunohistochemistry. Western blotting showed that MDA expression was significantly decreased in the bladder of the SHR–theanine group compared to the SHR–water group (*p* < 0.001) (Figure 5A). MDA was expressed in the urothelium and submucosal layer according to immunohistochemical staining. Expression of MDA in the bladder of the SHR–theanine group was decreased, as compared with the SHR–water group (Figure 5B), suggesting that l-theanine prevented bladder overactivity by reducing oxidative stress levels in the bladder of SHR.

### 2.5. Histological Changes in the Bladder Muscle Layer

HE staining was used to detect small arteries under the urothelium, which are thought to be responsible for bladder blood flow. The number of arterioles to be examined was about three in each rat sample. MLR of arteries was significantly lower in the SHR–theanine group than in the SHR–water group (*p* = 0.01) (Figure 6), indicating that l-theanine prevented chronic bladder ischemia by attenuating peripheral vascular resistance in SHR.

EM-stained bladder tissue showed that increases in connective tissue/muscle ratio in the muscle layer were significantly decreased in the SHR–theanine group, as compared with the SHR-water group (*p* = 0.02) (Figure 7).

## 3. Discussion

This study demonstrated that l-theanine might offer protective effects against bladder dysfunction. Bladder dysfunction can be divided into bladder storage dysfunction and voiding dysfunction. The mechanisms underlying bladder storage dysfunction in SHRs are related to prolonged sympathetic nerve hyperactivity, which is considered to directly affect both bladder storage function and chronic hypertension. For instance, activation of the sympathetic nervous system with aging elevates secretion of nerve growth factors by the bladder wall, causing bladder hyperactivity [17]. The bladder hyperactivity in SHR has also been suggested to involve noradrenergic control of the micturition reflex in the brain and spinal cord [19]. l-theanine significantly decreased the 24 h micturition frequency in SHRs in this study, particularly at night. Since rats are more active at night, serum noradrenaline levels are higher during the night than during the day [20]. Our study thus indicated that decreased serum noradrenaline levels in SHR may have contributed to protecting bladder storage function from bladder hyperactivity, especially during active times of the day.

Chronic hypertension in SHRs is also suggested to indirectly affect bladder dysfunction. Several studies have reported the antihypertensive effects of l-theanine. Although previous studies examined blood pressure immediately after single intraperitoneal injection [21] or single oral administration of l-theanine [6], the antihypertensive effects of long-term l-theanine solution intake have remained unclear. Investigating the effects of long-term oral administration of l-theanine is important for clinical practice because oral intake is much easier to incorporate into daily life. Oral administration of l-theanine for 6 weeks significantly decreased MBP, although SBP and DBP were not significantly decreased in the present study. The amount of l-theanine administered in this study was smaller than in previous studies [6,21]. This may be one reason why l-theanine had little effect on SBP and DBP. MBP depends on the blood vessel resistance in the small arteries, which are responsible for providing a steady supply of blood to peripheral organs, including the bladder [22]. The decreased MLR in suburothelial micro-arterioles in SHRs treated with l-theanine also suggests that blood flow to the bladder was preserved by reduced vascular resistance. Moreover, serum noradrenaline is well known to increase vascular resistance through its vasoconstrictive properties. Sympathetic small arteries in rodents are located in the bladder suburothelium, regulating blood flow in the bladder [23]. In this study, l-theanine significantly lowered only noradrenaline, not dopamine or adrenaline, suggesting that l-theanine may have been involved in the noradrenaline metabolic pathway. Our study indicated that long-term administration of l-theanine ameliorated excessive contraction of vascular smooth muscle in peripheral small arteries by suppressing elevated serum levels of noradrenaline in SHRs, thereby decreasing MBP and MLR.

Chronic hypertension causes microvascular alterations, leading to disturbed tissue perfusion of peripheral organs [24]. Since chronic bladder ischemia is well known as the cause of bladder dysfunction mediated by oxidative stress [25], a similar mechanism has been suggested in hypertension [26,27]. In this study, l-theanine suppressed bladder levels of the non-specific oxidative marker MDA, especially in the urothelium. The urothelium plays an important role in bladder storage function as a secretory organ [28]. Reductions in oxidative stress in the urothelium by l-theanine might thus contribute to protection from bladder storage dysfunction, including bladder hyperactivity in SHRs. In addition, chronic bladder ischemia impairs bladder contractility [29]. In this study, l-theanine inhibited the impaired bladder contractility and progression of fibrosis in the bladder muscle layer in SHRs, suggesting that increased blood perfusion in the bladder may also have protected bladder voiding function, along with storage function.

The concentration of l-theanine in this study was adjusted to that of commercial tea (approximately 2.7 mg/100 mg). The daily intake of l-theanine in this study was thus 0.46 mg/kg at the human-equivalent dose, meaning that a 60 kg human would have ingested 27.8 mg/day of l-theanine. Recently, supplements containing l-theanine have become widely available, with dosage forms ranging from 100 to 500 mg per tablet. In other words, approximately 1000 mL of green tea or one tablet of l-theanine per day is likely to help prevent hypertension-related bladder dysfunction.

Several limitations need to be considered when interpreting the findings from this study. Firstly, our study examined the effects of l-theanine in SHRs alone. One reason why we did not compare SHRs with rats without hypertension in this study is that SHRs are commonly used as an animal model of frequent urination and bladder dysfunction [26]. Another reason is that the appropriate controls for SHRs are controversial. Since SHRs were established by crossing WKY rats with spontaneously hypertensive rats, WKY rats are often used as controls for SHRs. However, some studies have questioned the use of WKY rats as controls for SHRs due to the complicated breeding history and large genetic divergence between WKY rats and SHRs [30,31]. Therefore, our study could not completely exclude the possibility that the SHRs used as controls did not develop bladder dysfunction. Secondly, the parasympathetic nervous system was not assessed in this study. Since bladder contractility is strongly reliant on acetylcholine and ATP secreted by parasympathetic nerve endings, it is possible that the parasympathetic nervous system was also involved in the preventive effects of l-theanine on lower urinary tract dysfunction. Thirdly, our study did not evaluate the effects of noradrenaline and adrenaline on the overall activity of SHRs or on the urethral tone. Since SHRs have also been used as an animal model for attention-deficit/hyperactivity disorder, which is known to be associated with LUTS [32], high noradrenaline levels in SHRs may have affected the overall activity related to urination. In addition, the urethral tone may be increased via postjunctional alpha adrenoceptors under sympathetic activation [33]. Finally, only male rats were used in this study in order to the experimental condition as uniform as possible. Bladder dysfunction in SHRs does not necessarily replicate bladder dysfunction associated with chronic hypertension in humans. However, we believe that our study has demonstrated some potentially preventive effects of l-theanine on bladder dysfunction associated with chronic hypertension.

Long-term oral administration of l-theanine did not show a significant reduction in SBP but may be contributed to the prevention of bladder dysfunction by reducing peripheral vascular resistance in this study. Our study represents an important contribution because oral l-theanine doses can be easily added to daily life to prevent bladder dysfunction, even taking these limitations into consideration. Further research on l-theanine is needed to clarify the mechanisms of bladder dysfunction and its effects in humans.

## 4. Materials and Methods

### 4.1. Animals and Experimental Design

Twenty 12-week-old male SHRs (SHR/Izm; Japan SLC, Shizuoka, Japan) were used in this study. Rats were placed under controlled conditions with a 12 h light/dark cycle and had free access to standard food pellets. Rats were divided into an SHR–water group and an SHR–theanine group (*n* = 10 each). The SHR–water group was given only water to drink, and the SHR–theanine group was given only l-theanine solution to drink, with both groups having free access. The l-theanine solution was created by dissolving l-theanine powder (APE8411, Wako, Japan) in water. The concentration of l-theanine solution was adjusted to 27 µg/mL, based on levels found in popular green tea-based beverages. Fluid intake was measured every day at 2:00 p.m., and drinking fluids were changed once every 2 days. After 6 weeks, heart rate (HR) and blood pressure parameters of systolic blood pressure (SBP), diastolic blood pressure (DBP), and mean blood pressure (MBP) were measured using the tail-cuff method (BP-98-L; Softron, Tokyo, Japan) without anesthesia. Subsequently, 24 h metabolic cage experiments and conscious free-moving cystometry were performed. After recording cystometrograms, rats were anesthetized with 2% isoflurane. Blood samples were collected by cardiac puncture, rats were sacrificed by cervical dislocation, then bladders were rapidly harvested and weighed. The bladder was opened caudally, and the trigone and dome were removed to exclude excessively damaged bladder sections. From the rest of the bladder, four rectangular strips with mucosa (approximately 5 × 2 mm) were cut longitudinally for pharmacological examinations. Other bladder regions were used for histological studies and Western blotting to analyze levels of oxidative stress. All protocols for these animal experiments were reviewed and approved by the Animal Ethics Committee at Fukushima Medical University (approval no. 2019011).

### 4.2. Metabolic Cage Experiments

Individual rats were placed in separate metabolic cages. These cages were attached to a urine collection funnel, which was placed over an electronic balance (A&D, Tokyo, Japan) to measure micturition behaviors. The amount of urine at each void was monitored using a multiport controller (PowerLab 4/26; AD Instruments, Dunedin, New Zealand). The first 48 h were considered an acclimatization period, then micturition behaviors in the last 24 h were used for analysis. Between 7:00 a.m. to 7:00 p.m. was defined as daytime and between 7:00 p.m. to 7:00 a.m. the next day as nighttime based on a previous study [34]. In this study, micturition parameters including 24 h urine volume, micturition frequency, daytime frequency, nighttime frequency, single voided volume, and total urine output were measured.

### 4.3. Conscious Free-Moving Cystometry

After metabolic cage experiments, cystometry was performed in conscious, free-moving rats as described previously from our laboratory [35]. A polyethylene catheter was implanted into the bladder dome and anchored with a 7-zero polypropylene suture for continuous measurement of bladder pressure. The other end of the catheter was passed subcutaneously through the back and out of the neck. After a 3-day postoperative recovery period, the rat was placed in a metabolic cage without any restraint or anesthesia. The bladder catheter was connected to a pressure transducer (AP-601G; Nihon Kohden, Tokyo, Japan) and a microinjection pump (STC-521; Terumo Co., Tokyo, Japan). Saline was continuously infused into the bladder at a rate of 12 mL/h. The same electronic balance with the urine collection system was applied as in the metabolic cage experiments. To stabilize the voiding pattern, rats were placed in the metabolic cage for at least 30 min. The following cystometric parameters were evaluated: intercontractile interval, bladder capacity, voided volume, residual volume, maximal pressure, and bladder compliance.

### 4.4. Organ Bath Study

All chemicals used in the organ bath study were purchased from Wako Pure Chemical Industries (Tokyo, Japan). The rectangular bladder strips were suspended in a 25 mL organ bath containing Krebs solution. The measurement of bladder isometric tension was carried out using a force–displacement transducer (TB-621; Nihon Kohden, Tokyo, Japan). Strips were subjected to 1 g of resting tension and allowed to stabilize for at least 60 min. After stabilization, changes in strip tension were recorded from baseline in response to 80 mm of KCl, electric field stimulation (EFS) (1, 2, 4, 8, 16, or 32 Hz), 1 mm of adenosine triphosphate (ATP), and carbachol (concentration from 1 nm to 1 mm) in turn. All responses were normalized to 1 g tissue weight.

### 4.5. Detection of Oxidative Stress Markers

The protein expression level of the oxidative stress marker malondialdehyde (MDA) in the bladder was quantified by Western blotting, as follows. Frozen bladder tissues were homogenized and dissolved in a buffer containing 8 m urea and 10 mm dithiothreitol to extract proteins. Total protein concentrations were measured using a NanoDrop Lite UV–Vis Spectrophotometer (Thermo Fisher Scientific, Waltham, MA, USA). Samples were then diluted with 5× sodium dodecyl sulfate (SDS) buffer and boiled for 3 min. Each sample (10 µg of protein per lane) was subsequently subjected to SDS–Polyacrylamide gel electrophoresis and blotted onto a polyvinylidene difluoride membrane. Next, 1% polyvinylpyrrolidone in TBS-T (20 mm Tris pH 7.5, 0.5 m NaCl, 0.1% Tween 20) was used to block the membranes. The rabbit polyclonal anti-MDA (SMC-514; Stressmarq Biosciences, Victoria, Canada) and mouse monoclonal anti-β-actin (A5316; Sigma-Aldrich, St. Louis, MO, USA) were used in the experiment as primary antibodies. The secondary antibodies were horseradish peroxidase (HRP)-conjugated antibodies appropriate for each primary antibody. Target proteins were colorized with SuperSignal West Dura Extended Duration Substrate (Thermo Fisher Scientific, Waltham, MA, UAS) and imaged with a ChemiDoc XRS plus system (BIO-RAD, Hercules, CA, USA). The β-actin was evaluated as a loading control to ensure that the amounts of protein loaded on each lane were comparable.

### 4.6. Histological Examination

Fragments of bladders fixed in 10% neutral-buffered formalin were embedded in paraffin. Bladder tissue was cut at a thickness of 3 µm for hematoxylin and eosin (HE) and Elastica Masson (EM) staining, and at a thickness of 5 µm for immunohistochemical staining.

To investigate minimal vascular resistance, HE-stained arterioles in the bladder submucosa were evaluated using the media-to-lumen ratio (MLR) [36]. The arterioles to be evaluated are limited to circles less than 100 µm in diameter between the urothelium and bladder muscle layer [37].

Computer-assisted imaging histomorphometric analysis of the EM-stained bladder wall was performed using cellSens Dimension version 2.3 software (OLYMPUS, Tokyo, Japan). The ratios of smooth muscle area (red stained) to connective tissue area (blue stained) in four randomly selected full-thickness sites in the bladder muscle layers were calculated.

For immunohistochemical examination of bladder tissue, sections were deparaffinized and endogenous peroxidase was inactivated with 0.3% H_2_O_2_. We used 5% skim milk to block nonspecific immunoglobulin G binding. Bladder sections were incubated overnight at 4 °C with primary antibodies for MDA (1:500, SMC-514; Stressmarq Biosciences), then incubated for 1 h at room temperature with HRP-conjugated secondary antibodies appropriate for the primary antibody (diluted, Signal Stain^®^ Boast IHC Detection Reagent; Cell Signaling Technology, Tokyo, Japan). Localization of MDA-positive cells was evaluated in bladder tissues.

### 4.7. Statistical Analysis

Data were analyzed using IBM SPSS Statistics version 27 software (Statistical Package for the Social Sciences, Chicago, IL). Characteristics of the study samples are presented as mean ± standard deviation (SD). Comparisons between groups were performed using the Mann–Whitney test. Phasic and tonic contractions in each muscle strip were compared using the paired *t*-test. Values of *p* < 0.05 were considered significant.

## 5. Conclusions

Figure 8 summarizes a potential hypothesized pathway from our findings regarding the preventive effects of l-theanine for bladder dysfunction, including bladder storage and voiding dysfunction, directly or indirectly through suppression of chronically high levels of noradrenaline. Since the dose of l-theanine used in this study was suitable for human consumption, further findings on the optimal effects of l-theanine in preventing bladder dysfunction with chronic hypertension are expected.

## Figures and Tables

**Figure 1 metabolites-11-00778-f001:**
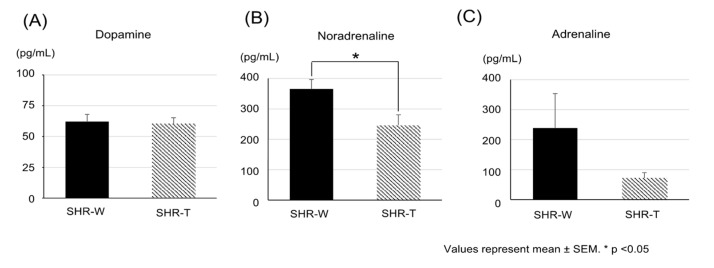
Mean serum catecholamine concentrations after 6 weeks of administration: (**A**) serum dopamine concentration, (**B**) serum noradrenaline concentration, and (**C**) serum adrenaline concentration of rats assigned to the two groups. Serum noradrenaline concentration was significantly lower in the SHR–theanine group than in the SHR–water group. Data are presented as mean ± standard error of the mean. * Statistical significance at the *p* < 0.05 level.

**Figure 2 metabolites-11-00778-f002:**
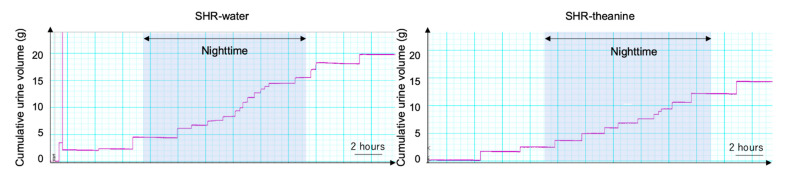
Typical 24 h voiding behavior by metabolic cage in the SHR–water and SHR–theanine groups. Gray shading represents nighttime. Scale bars = 2 h.

**Figure 3 metabolites-11-00778-f003:**
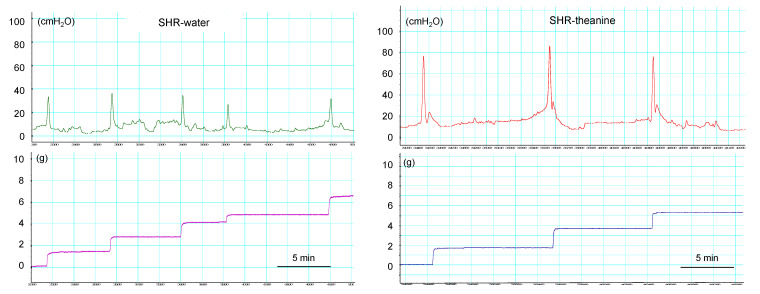
Typical cystometrograms in the SHR–water and SHR–theanine groups. Micturition interval is significantly longer, and voided volume is significantly larger in the SHR-theanine group than in the SHR–water group. Scale bars = 5 min.

**Figure 4 metabolites-11-00778-f004:**
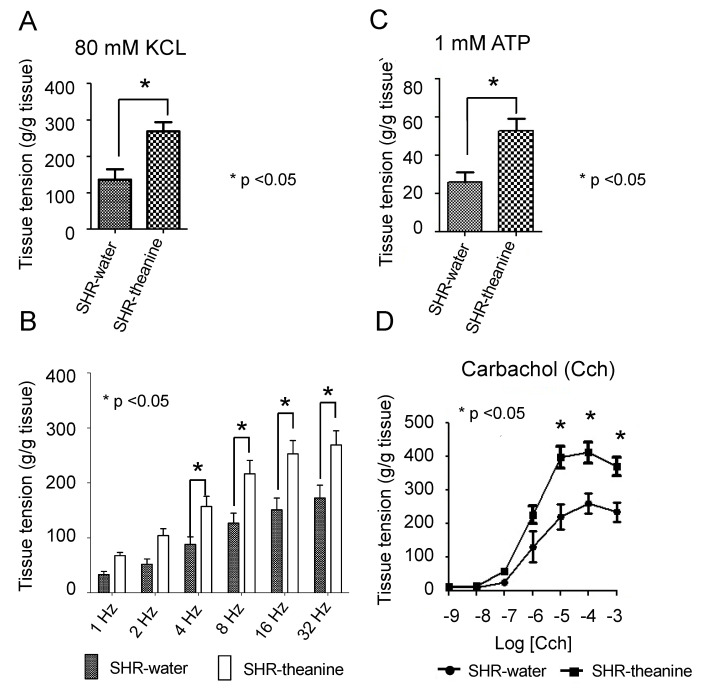
Comparison of bladder strip contractility in response to various stimuli: (**A**) comparison of bladder strip contractility in response to KCl, (**B**) electrical field stimulation, (**C**) ATP, and (**D**) carbachol between SHR treated with water and treated with l-theanine. Mean contractile responses to 80 mm KCL or 1 mm ATP, electrical field stimulation from 4 Hz to 32 Hz, or carbachol concentration from 10 μM to 1 mM were significantly larger in the SHR–theanine group than in the SHR–water group. Data are presented as mean ± standard error of the mean. * Statistical significance at the *p* < 0.05 level.

**Figure 5 metabolites-11-00778-f005:**
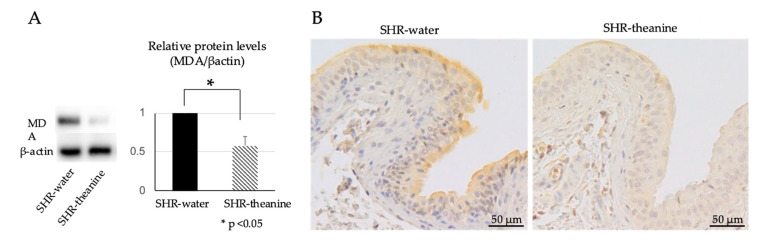
Summary of bladder oxidative levels in SHR treated with water and treated with l-theanine: (**A**) relative MDA protein levels in bladder and Western blots. Protein expression level of the bladder was significantly lower in the SHR–theanine group than in the SHR–water group. * Statistical significance at the *p* < 0.05 level; (**B**) immunohistochemical staining of the bladder in SHR–water and SHR–theanine groups, scale bars = 200 μm. Expression of MDA in the bladder was weaker in the SHR–theanine group than in the SHR–water group.

**Figure 6 metabolites-11-00778-f006:**
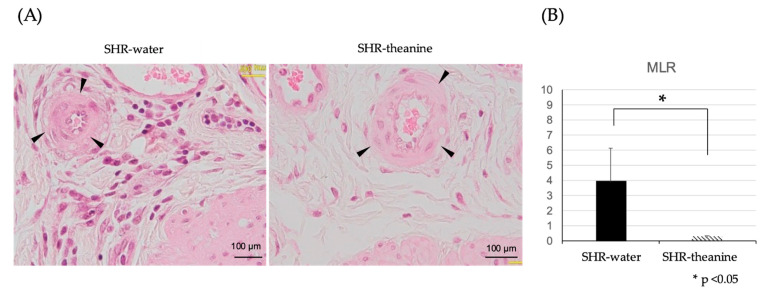
Media-to-lumen ratio of bladder small arteries the SHR-water and SHR-theanine groups: (**A**) hematoxylin and eosin staining of suburothelial arterioles (arrowheads) in SHR treated with water and treated with l-theanine, scale bars = 100 μM; (**B**) the MLR is significantly lower in the SHR–theanine group than in the SHR–water group (**B**). * Statistical significance at the *p* < 0.05 level.

**Figure 7 metabolites-11-00778-f007:**
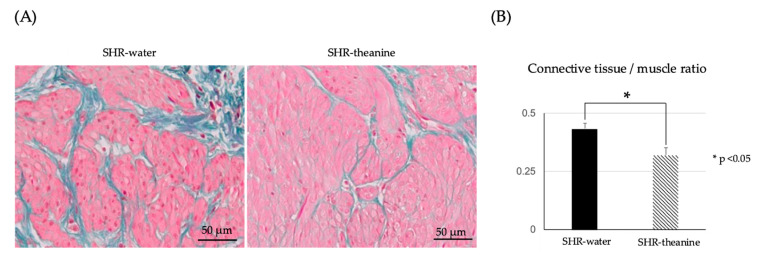
Bladder smooth muscle to connective tissue ratio in the SHR-water and SHR-theanine groups: (**A**) Elastica–Masson staining of the detrusor smooth muscle layer of the bladder in SHR treated with water and treated with l-theanine. Scale bars = 50 μM; (**B**) the connective tissue-to-muscle ratio is significantly lower in the SHR–theanine group than in the SHR–water group. * Statistical significance at the *p* < 0.05 level.

**Figure 8 metabolites-11-00778-f008:**
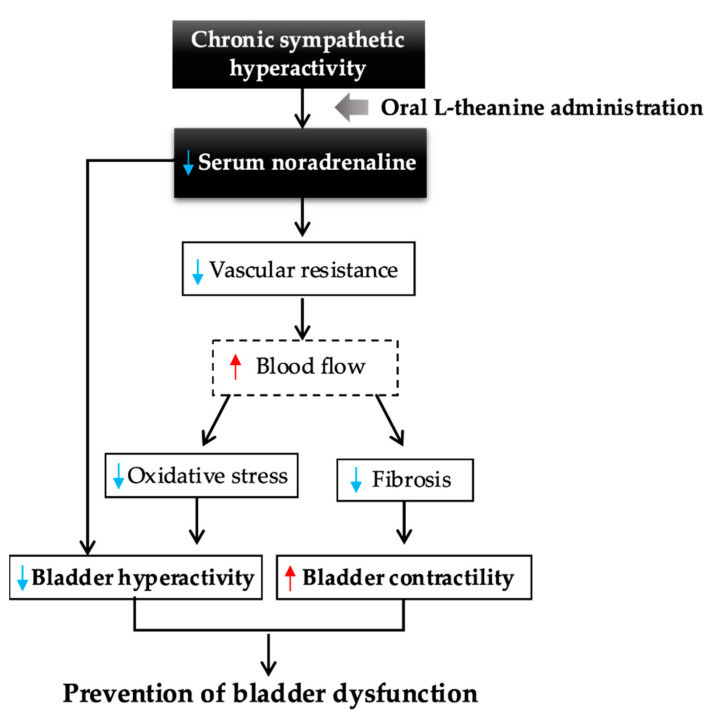
Summary of a potential hypothesized pathway for preventive effects of l-theanine on bladder dysfunction. Decreased serum noradrenaline level by l-theanine administration directly prevents bladder dysfunction with bladder hyperactivity. As an indirect pathway, decreased vascular resistance inhibits the increase in oxidative stress and fibrosis associated with bladder ischemia, resulting in suppression of bladder hyperactivity and prevention of impaired bladder contractility.

**Table 1 metabolites-11-00778-t001:** General features of SHRs in the SHR–water and SHR–theanine groups. * Statistical significance at the *p* < 0.05 level.

	SHR-Water			SHR-Theanine			*p* Value
Fluid intake (mL/day)	33.9	±	2.8	33.8	±	1.7	1.000
Body weight (g)	353.0	±	12.3	354.0	±	11.3	0.912
Bladder weight (g)	0.19	±	0.03	0.22	±	0.04	0.515
HR (beats/min)	412.3	±	18.5	391.9	±	20.7	0.606
SBP (mmHg)	204.0	±	5.8	188.8	±	3.8	0.074
MBP (mmHg)	173.3	±	4.3	159.9	±	3.3	0.046 *
DBP (mmHg)	159.3	±	3.9	147.9	±	4.1	0.074

**Table 2 metabolites-11-00778-t002:** Comparison of each parameter in 24 h metabolic cage experiments. * Statistical significance at the *p* < 0.05 level.

	SHR-Water			SHR-Theanine			*p* Value
Urine volume (g/24-h)	14.4	±	3.3	12.9	±	3.2	0.393
Micturition frequency (count/24-h)	22.3	±	5.3	16.3	±	4.0	0.004 *
Daytime frequency (count/12-h)	5.0	±	1.6	5.1	±	1.2	0.796
Nighttime frequency (count/12-h)	16.3	±	4.3	11.2	±	3.6	0.007 *
Single voided volume (g)	0.67	±	0.19	0.81	±	0.15	0.043 *

**Table 3 metabolites-11-00778-t003:** Cystometory parameters in the SHR–water and SHR–theanine groups. * Statistical significance at the *p* < 0.05 level.

	SHR-Water			SHR-Theanine			*p* Value
Intercontractile interval (min)	5.6	±	8.4	7.8	±	1.7	0.002 *
Baseline pressure (cmH_2_O)	9.59	±	5.56	8.93	±	4.83	0.739
Voided volume (g)	0.96	±	0.18	1.34	±	0.35	0.029 *
Residual urine volume (mL)	0.16	±	0.08	0.22	±	0.13	0.247
Maximum pressure (cmH_2_O)	47.1	±	16.9	70.4	±	33.1	0.123
Bladder compliance (g/cmH_2_O)	0.12	±	0.79	0.11	±	0.03	0.393

## Data Availability

The data presented in this study are available in the article.

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
