# Peer review of "l-Theanine Protects Bladder Function by Suppressing Chronic Sympathetic Hyperactivity in Spontaneously Hypertensive Rat"

_metabolites, 2021, doi:10.3390/metabo11110778_

Round 1
Reviewer 1 Report
The authors use the spontaneous hypertensive rat (SHR) to study effects of L-theanine in preventing deleterious effects on bladder function. SHR rats receiving the L-theanine had decreased urine frequency, increased void intervals, bladder capacity and void volume compared to vehicle SHR rats. L-theanine treatment also resulted in increased bladder contractility in response to KCl, EFS, carbachol and ATP. Levels of noradrenaline were also lower along with MLR of blood vessels, and connective tissue compared to controls. While this is an important area of study, there are several concerns which are listed below:
- Bladder contractility is also heavily reliant on the parasympathetic nervous system, and the organ bath studies using carbachol and ATP target this pathway as they activate cholinergic and purinergic signaling respectively. At the very least there should be mention in the discussion of the importance of the parasympathetic pathway. In line with this - mention of the L-theonine acting purely through the sympathetic pathway should be dampened.
- There are several areas in the results and the discussion where the wording should be altered to not overstep the conclusions that can be drawn from the results. For example Page 3 ;line 87: “These results indicate that L-theanine prevented bladder hyperactivity in SHR”. There is nothing to compare and say what is hyperactive or not, the only thing that can be concluded is that L-theanine increased IVI, capacity and void volume compared to control SHR rats. Similarly Page 4; line105. “…Prevents declines in bladder contractile response”. Without comparing to a non SHR rat this should be rephrased and focus only on the effects of L-theanine observed here in SHR rats.
- Urethral pathways may also be involved and discussion of the sympathetic nervous system and urethral tone could be added to the discussion.
- Discussion could be added on the overall effects on activity of these animals, do they tend to be less active due to lower noradrenaline and adrenaline? Could these effects contribute to the bladder effects as well.
- Figure 4B would need quantification before the statement in the results (Page 5, line 113-115) should be made. No statements regarding the location of the reduction in MDA can be made unless the IHC is quantified, or unless westerns are preformed on separated urothelium.
- Figure legends could contain more detail in terms of endpoint examined, animal number and what arrows are indicating etc, to aid in readability. Also labels/units for Y-axis are missing in some of the panels.
- If cystometry was performed on the same animals used for MDA western or IHC were there any measures in place to confirm that the bladder did not sustain too much damage to be used for these endpoints?
8.Dilutions of antibodies used should be added to the methods.
- More detail is needed into the n values for each endpoint, were 10 animals used for each parameter including for western blots and immunohistochemistry and bladder baths?
- A sentence describing why only male rats were used would be beneficial.
- The final Figure 7 should indicate that this is a potential hypothesized pathway. This study did not mechanistically link any of these parameters to the treatment. Therefore this is a proposed possible pathway which needs further study.
Reviewer 2 Report
The authors ought to be praised for what appears to be an impecable experiment, with results which could be of interest for dietary adaptations or herbal medicine applications
Author Response
Thank you very much for your comments.
Reviewer 3 Report
This manuscript investigated whether L-theanine, a major amino acid found in green tea, could ameliorate bladder dysfunction induced by chronic sympathetic hyperactivity. The authors used SHRs, as an animal model of bladder dysfunction, and divided into SHR-water and SHR- theanine groups. After 6 weeks of oral administration of L-theanine, the bladder was taken for evaluation. Results showed nicturition interval was significantly longer and bladder capacity was significantly higher in the SHR-theanine than in the SHR-water group. Bladder strip contractility was also higher in the SHR-theanine than in the SHR-water group. Western blotting showed that expression of malondialdehyde was significantly lower in the SHR-theanine group. The authors concluded that L-theanine can prevent the progression of bladder dysfunctions by inhibiting chronic sympathetic hyperactivity and protecting bladder dysfunction.
Overall, this is an interesting article.
Here are my concerns and comments:
- The metabolic cage study showed daytime frequency is not different between groups, but nighttime frequency is significantly higher in the SHR-water group. Is there any good explanation?
- Please define Day-time frequency and Night-time frequency. Why used count/24-h to present your data?
- Please provide tracing analysis picture of 24 h voiding behavior by metabolic cage.
- The maximum pressure is much higher in SHR-theanine group (70.4±1 vs 47.1± 16.9 cmH2O), however, the picture below showed the maximum pressure in SHR-theanine group is even lower than the SHR-water group.
- It is hard to compare media to lumen ration in different arteries. How many arteries did you calculate in each rat sample?
- The saline infusion rate for conscious cystometry is 12ml/Hour, which is really high. Please provide the 30 minutes cystometry tracing recording.
- Is there any difference to make tea in cold or hot water to get equivalent L-theanine?
- The study needs another control group without hypertension.
Reviewer 4 Report
Remove: 'can prevent the progression' (this is not shown)
Add: it is unclear how similar the rats (tension) were at baseline (what is the usual range of BP in these rats) but better: provide baseline-data).
Explain the difference in 'bladder capacity (tab3)' and 'single voided volume (tab2)'
Explain that inhibition of sympathetic hyperactivity had little effect on BP.
Explain how 'disfunctional' the Tab2 and Tab3' are (compared to normal rats)
Maybe discuss the lack of vidence for hypoxia (or medication to reduce hypoxia) in human for LUTD. 'bladder dysfunction .... is suggested' (P1L41-42), I agree, suggested without any clinical evidence....
Round 2
Reviewer 1 Report
The authors have addressed all comments.
Reviewer 3 Report
The manuscript had been revised properly. I suggest to accept the manuscript in the current form.
Reviewer 4 Report
No further questions or comments